# The Triple Amino Acid Substitution TAP-IVS in the *EPSPS* Gene Confers High Glyphosate Resistance to the Superweed *Amaranthus hybridus*

**DOI:** 10.3390/ijms20102396

**Published:** 2019-05-15

**Authors:** Maria J. García, Candelario Palma-Bautista, Antonia M. Rojano-Delgado, Enzo Bracamonte, João Portugal, Ricardo Alcántara-de la Cruz, Rafael De Prado

**Affiliations:** 1Department of Botany, Ecology and Plant Physiology, University of Cordoba, 14071 Córdoba, Spain; b92gadem@uco.es (M.J.G.); qe2pabac@uco.es (C.P.-B.); 2Department of Agricultural Chemistry and Edaphology, University of Cordoba, 14071 Cordoba, Spain; q92rodea@uco.es (A.M.R.-D.); qe1pramr@uco.es (R.D.P.); 3Faculty of Agricultural Sciences, National University of Cordoba (UNC), 5001 Cordoba, Argentina; ebracamo@gmail.com; 4Department of Biosciences, Research Center for Endogenous Resource Valorization’s, Polytechnic Institute of Beja, 7800-295 Beja, Portugal; jportugal@ipbeja.pt; 5Departamento de Química, Universidade Federal de São Carlos, 13565-905 São Carlos, Brazil

**Keywords:** 5-enolpyruvylshikimate-3-phosphate synthase, EPSPS gene mutation, glyphosate-resistant crops, nontarget site, smooth pigweed, target site resistance

## Abstract

The introduction of glyphosate-resistant (GR) crops revolutionized weed management; however, the improper use of this technology has selected for a wide range of weeds resistant to glyphosate, referred to as superweeds. We characterized the high glyphosate resistance level of an *Amaranthus hybridus* population (GRH)—a superweed collected in a GR-soybean field from Cordoba, Argentina—as well as the resistance mechanisms that govern it in comparison to a susceptible population (GSH). The GRH population was 100.6 times more resistant than the GSH population. Reduced absorption and metabolism of glyphosate, as well as gene duplication of 5-enolpyruvylshikimate-3-phosphate synthase (EPSPS) or its overexpression did not contribute to this resistance. However, GSH plants translocated at least 10% more ^14^C-glyphosate to the rest of the plant and roots than GRH plants at 9 h after treatment. In addition, a novel triple amino acid substitution from TAP (wild type, GSH) to IVS (triple mutant, GRH) was identified in the EPSPS gene of the GRH. The nucleotide substitutions consisted of ATA^102^, GTC^103^ and TCA^106^ instead of ACA^102^, GCG^103^, and CCA^106^, respectively. The hydrogen bond distances between Gly-101 and Arg-105 positions increased from 2.89 Å (wild type) to 2.93 Å (triple-mutant) according to the EPSPS structural modeling. These results support that the high level of glyphosate resistance of the GRH *A. hybridus* population was mainly governed by the triple mutation TAP-IVS found of the EPSPS target site, but the impaired translocation of herbicide also contributed in this resistance.

## 1. Introduction

Plant breeding methods have delivered herbicide resistant crops that offer advantages for weed control [1]. The introduction of glyphosate-resistant (GR) crops in 1996 revolutionized weed management practices [2]. Agricultural areas occupied by these crops, mainly GR-soybean and GR-corn, increased considerably in Argentina, Brazil and USA [2,3]. Inadequate adoption of GR-crops, i.e., higher doses and more applications of glyphosate than recommended by the manufacturer, has selected for a wide range of superweeds (GR-weeds selected in GR-cropping systems), decreasing the value of this technology [4,5,6]. *Lolium rigidum* was the first weed characterized as being resistant to glyphosate, also in 1996 [7]. Since then, 43 weed species with glyphosate resistance have been reported worldwide [7].

Glyphosate (N-(phosphonomethyl) glycine) is applied for weed control in disturbed and undisturbed areas, since it is a broad-spectrum, nonselective and systemic herbicide [8,9]. Glyphosate inhibits the 5-enolpyruvylshikimate-3-phosphate synthase (EPSPS) disturbing the shikimate pathway, which prevents the biosynthesis of aromatic amino acids (phenylalanine, tyrosine and tryptophan) [8,9,10]. Thus, low shikimate accumulation in resistant plants compared to susceptible ones is an indicator of resistance [11]. That resistance may be monogenic or polygenic [12]. Monogenic resistance is governed by the gene related to the target site via gene mutations or gene amplification, referred to as target site resistance (TSR) mechanisms [8,10,13]. Polygenic resistance is regulated by several genes nonrelated to the target site gene, responsible for the reduction of glyphosate absorption and/or translocation, glyphosate metabolism, sequestration of the herbicide into vacuole, or foliar hypersensitivity, referred to as nontarget site resistance (NTSR) mechanisms [12,14,15,16]. These divergent mechanisms show how the plants adapt to the xenobiotic stress exerted by the herbicides, selecting for resistance [17].

Several traits make *Amaranthus* a dominant and difficult weed genus to control in summer crops, such as: its high growth rate, high fecundity, great genetic variability, tolerance to stress and the ability to select for herbicide resistance [18,19]. *Amaranthus hybridus* and *A. palmeri*, both named as “yuyo colorado” by Argentinian farmers due to the difficulty of distinguishing them morphologically, are two predominant weed species in Argentinian agricultural areas used in the production of GR-crops [20]. Glyphosate efficiently controlled *Amaranthus* species, but recently, a lack in control of *A. hybridus* was observed in GR-soybean fields in South of Cordoba, Argentina [20].

Resistance mechanism characterization should not be ignored, since implementing weed management measures without characterizing them may aggravate resistance from monogenic to polygenic, as reported in *A. tuberculatus* var. *rudis*, *Epilobium ciliatum*, and *Kochia scoparia* that selected resistance for six, five, and four modes-of-action of herbicides, respectively [21,22,23]. Mechanisms conferring glyphosate resistance were characterized in *A. palmeri*, *A. spinosus* and *A. tuberculatus* [10,24,25]; but there are no studies for *A. hybridus*. Here, an *Amaranthus* sp. population (GRH) with a high glyphosate resistance level, collected in a GR-soybean field from Cordoba (Argentina) and carrying a triple amino acid substitution in the conserved region of the *EPSPS* gene [26], was confirmed as being *A. hybridus* by genotyping and their TSR and NTSR mechanisms conferring it glyphosate resistance were characterized in comparison with a susceptible population (GSH).

## 2. Results

### 2.1. Shikimic Acid Accumulation and Genotyping

Shikimic acid accumulation differentiated the susceptibility to glyphosate of the GSH and GRH populations. GSH plants presented different shikimate accumulation patterns, while the GRH population seems to be more homogeneous (Figure 1A). Thus, GSH (eight) and GRH (nine) plants, with similar shikimate accumulation profiles, were selected to obtain homogeneous lines for future experiments. Ten F_1_ plants from each population were confirmed as being *A. hybridus* (Figure 1B).

### 2.2. Dose–response Assays

Plant survival and fresh weight decreased as the dose of glyphosate increased (Figure 2). The GSH population died at lower glyphosate doses than the doses used by Argentinian farmers (960 g ae ha^−1^). The GR_50_ and LD_50_ values estimated for the GRH population were 1395.2 and 3503.4 g ae ha^−1^ glyphosate, respectively. Looking at the LD_50_ parameter, the RI for GRH was 100.6 times more resistant than the GRH population (Table 1).

### 2.3. ^14^C-Glyphosate Absorption and Translocation

The ^14^C-glyphosate absorption ranged between 12 and 76% from 12 to 96 HAT, but there were no differences between GRH and GSH plants (Figure 3A). Already, the GSH plants moved more ^14^C (glyphosate or metabolites) from the treated leaf to the remainder of plant and roots than GRH plants (Figure 3B). Quantitative translocation results showed that GRH plants retained ~68% of the glyphosate in the treated leaf while the GSH plants only retained ~53%. Thus, the ^14^C-herbicide translocated to the rest of the plant and roots was up to ~18 and ~28%, respectively, in GSH plants, while in GRH plants it was ~14 and ~18% (Figure 3C).

### 2.4. Glyphosate Metabolism

Glyphosate was not metabolized by *A. hybridus* plants. At 48 and 96 HAT, GRH plants translocated less herbicide towards roots than GSH ones, observing the biggest differences at 96 HAT. The amounts of glyphosate found in the *A. hybridus* plants ranged from 437 to 605 nmols g^−1^ fresh weight in the aerial part, and from 48 to 238 nmols g^−1^ fresh weight in roots (Table 2). Small amounts of glyoxylate were detected at 96 HAT in both GSH and GRH plants (Figure 4), but this compound cannot be considered a metabolite because it does not originate solely from glyphosate. This data confirmed the results obtained in the absorption and translocation assays, i.e., the ^14^C detected by LSS corresponded to glyphosate molecules and not a possible metabolite.

### 2.5. EPSPS Enzyme Activity Assays

*Amaranthus hybridus* plants (GSH and GRH) presented similar EPSPS basal activity profiles (Figure 5A). The EPSPS was inhibited in both GSH and GRH plants as glyphosate concentrations increased, but inhibition occurred at different herbicide concentrations (Figure 5B). These concentrations were 0.52 and 52.8 µM glyphosate for the GSH and GRH populations, respectively, resulting in an RI (R-to-S ratio) of 100.9.

### 2.6. EPSPS Copy Number and Gene Expression

EPSPS copy gene number relative to ALS gene showed that there were no differences between GSH and GRH biotypes. However, a slight increase in the *EPSPS* gene expression was observed (Figure 6). The EPSPS and expression copy number averages were 0.99 ± 0.03 and 0.82 ± 0.04, respectively, for the GSH population, and 1.36 ± 0.05 and 1.45 ± 0.10 for the GRH population.

### 2.7. Mutations in the EPSPS Coding Sequence

EPSPS gene sequencing revealed a novel amino acid substitution from Ala to Val at the 103 position in the GRH population, in addition to those already described at Thr-102-Ile and Pro-106-Ser, i.e., a triple mutation was found. This triple amino acid substitution consisted of TAP (wild type, GSH) to IVS (triple mutant, GRH). The nucleotide substitutions were ATA^102^, GTC^103^, and TCA^106^ instead of ACA^102^, GCG^103^, and CCA^106^, respectively (Figure 7A). The *A. hybridus* EPSPS cDNA sequences can be found in GenBank (accession numbers MG595170-MG595171).

### 2.8. Structural Modeling

EPSPS structural modeling revealed increases in the hydrogen bond (H bond) distances between the Gly-101 and Arg-105 positions of the EPSPS gene from GRH plants. In EPSPS target site of GSH plants, the H bond distance between these two amino acids was 2.89 Å, while in GRH plants this distance was 2.93 Å (Figure 7B,C).

## 3. Discussion

Argentinian farmers quickly adopted the technology packages for cultivation of GR-crops, mainly soybean and corn [4]. In the present study, the degree of glyphosate resistance in a population of the superweed *A. hybridus*, occurring in a GR-soybean field from Cordoba, was characterized, as well as the resistance mechanisms that govern it. The accumulation of shikimic acid is a biochemical indicator of glyphosate effects on susceptible plants [11]. Once the glyphosate susceptibility profile of the GSH and GRH populations via shikimate accumulation were differentiated, F_1_ individuals from these populations were identified as being *A. hybridus* subsp. *hybridus*, based on the length of intron 1 of the EPSPS gene [27,28].

The dose–response assays showed that the GRH population had a high level of resistance to glyphosate. The GR_50_ estimated for the GRH populations was higher than the glyphosate dose of 960 g ae ha^−1^ used in the field by Argentinian farmers. However, this GR_50_ (1395 g ae ha^−1^) was 15 times less than the GR_50_ value determined for another R population of *A. hybridus* carrying the triple substitution TAP-IVS [29]. These differences were mainly due to the phenological stage to which the plants were treated in both experiment (4-true leaves vs 20 cm in height (8–12 true leaves)), since older plants are less susceptible to herbicides [30]. In addition, the volume applied and the equipment used for glyphosate applications (200 L ha^−1^ in a treatment chamber vs 75 L ha^−1^ with a backpack sprayer) influenced in the differential response observed in both resistant populations. However, the high level of glyphosate resistance selected by the two resistant populations of *A. hybridus*, which were collected in different regions of the province of Cordoba, Argentina, was consistent. The impact on crop yield of superweeds is strong due to the wide use of glyphosate and because GR crops are designed to tolerate this herbicide [4]. In GR crop systems from Argentina, *Shorgum halepense* was confirmed as being glyphosate-resistant [31], and other common weeds as tolerant [32]. 

Reduced absorption and/or metabolism of glyphosate were not relevant in the high resistance of the GRH *A. hybridus* population; however; the translocation was significantly lower than the GSH population. Previously, an unknown barrier in the phloem system or in the mesophyll cells was suggested as limiting glyphosate translocation [33]. Studies with ^31^P nuclear magnetic resonance have shown that glyphosate is compartmentalized and deactivated into the vacuole [14,34,35], a process that is regulated by tonoplast-active transporters [35]. This NTSR mechanism endows moderate-level glyphosate resistance [36]; therefore, the high resistance level of the GRH population (100.6-times in relation to the GSH) cannot be explained only by this mechanism alone, suggesting that TSR mechanism may be preventing the interaction of glyphosate with the EPSPS.

The GSH and GRH *A. hybridus* populations presented similar *EPSPS* gene copy numbers, and although there was a slight increase in the expression of this enzyme in the GRH population, such expression did not result in greater EPSPS basal activity. *EPSPS* gene amplification was reported in several GR *Amaranthus* species. For example, *A. palmeri* had from 5 to 160 copies of the *EPSPS* gene [10], *A. tuberculatus* had from 32 to 59 copies [37], and *A. spinosus* presented 33–37 copies [38]. Given that the GRH *A. hybridus* plants did not show an increased EPSPS gene copy number, the slight increase in *EPSPS* gene expression did not explain the differences observed in the EPSPS activity.

Interestingly, for the first time, an EPSPS triple mutation (Thr-Ile 102, Ala-Val 103, and Pro-Ser 106) was found in the GRH *A. hybridus* population [25], which could explain the high levels of glyphosate resistance observed in these plants. Several mutations in the *EPSPS* gene have been suggested as contributing to glyphosate resistance [8,39]. However, the mutations conferring resistance to this herbicide must occur in the conserved region of this gene (^95^LFLGNAGTAMRPL^107^), that interact directly with glyphosate, as demonstrated in *Escherichia coli* [40,41]. Single amino acid substitutions occurring at the 106 position from Pro to Ser, Ala, Thr, and/or Leu, endowing total or partial resistance to glyphosate, has been widely reported in mono- and dicotyledonous weeds [8]. These single mutations provide low levels of resistance (2–3-fold), but they are sufficient for weeds to survive at field doses of glyphosate [36]. Single Pro-106 substitutions cause a slight narrowing of the glyphosate/phosphoenolpyruvate (PEP) binding site cavity, endowing glyphosate resistance but preserving EPSPS functionality [40]. Single substitutions at Gly-101 and Thr-102 confer high-level glyphosate resistance, but decrease the volume of the glyphosate/PEP binding site, reducing affinity for PEP [41,42]. Sammons and Gaines [8] warned that Thr-102 mutations would be unlikely to occur first or independently of Pro-106 mutations due to their significant reduction in PEP Km; however, *Tridax procumbens* presented a novel single mutation from Thr to Ser at the EPSPS genomic position 102 [43]. The double mutation known as TIPS (Thr-102-Ile + Pro-106-Ser), used in some GR-crops, was reported in *Eleusine indica* [36,44] and *Bidens pilosa* [13], endowing them high glyphosate resistance levels (GR_50_ values ranging from 1055 to 2050 g ae ha^−1^). The GRH *A. hybridus* population showed an additional mutation, occurring at the 103 position, associated with the TIPS mutation, which could be responsible for further increasing the level of glyphosate resistance.

How the Ala-103-Val substitution can affect the protein function is still unknown, but based on the amino acid properties, the effect that this mutation could produce on the EPSPS activity can be predicted. The side chain of alanine is poorly reactive and rarely participates in protein function, but may play a role in the recognition or specificity of the substrate [45]. Being hydrophobic, valine prefers to be buried in hydrophobic protein cores; however, this amino acid has an often overlooked property, it is branched into Cβ like isoleucine and threonine [45]. While most amino acids contain only one nonhydrogen substituent on their Cβ carbon, these three amino acids contain two. This means that there is much more volume in the main chain of the protein, such that these amino acids are more restricted in the conformations of the main chain that they can adopt [45]. The substitution Val-103-Ala found in the GRH *A. hybridus* plants increased the distance between the H bonds in the α-helix of EPSPS target site, i.e., the alanine side chain occupies more space than the valine one. Therefore, the occurrence of the triple substitution in the conserved region of the *EPSPS* encoding gene supports that TAP-IVS mutation contributed to the high level of glyphosate resistance of the GRH *A. hybridus* population. In addition, in silico conformational studies in another glyphosate-resistant population of *A. hybridus*, also collected in the province of Cordoba Argentina, showed that the TAP-IVS mutation restricts the binding of glyphosate with the EPSPS enzyme [29].

## 4. Materials and Methods 

### 4.1. Plant Materials

Mature seeds of a resistant *A. hybridus* population (GRH) were harvested in no-till GR-soybean cropping systems from the Cordoba province (Argentina) (31.48° S, 64.01° W) in the crop season 2015/2016. This field had a 20-year history of glyphosate application (twice per crop season) at 960 g ae ha^−1^. There was one application (glyphosate alone or in mixture with a residual pre-emergent herbicide) prior to crop sowing, and another 30 d after sowing. In addition, this field was cultivated with wheat Clearfield^®^ in the winter season. Susceptible (GSH) seeds were collected in a nearby area with no history of herbicide application in 2016.

The GRH and GSH populations were germinated in trays containing peat and sand, moistened with distilled water and placed in a growth chamber (28/18 °C day/night, 16-h photoperiod, 850 µmol m^−2^ s^−1^ photosynthetic photon flux, and 80% relative humidity). Seedlings were then transplanted individually into 250 mL pots containing sand/peat in a 1:2 (*v*/*v*) ratio, and kept in the growth chamber until the herbicide applications. Afterwards, treated plants were placed in a greenhouse under similar conditions to those of the growth chamber.

### 4.2. Fast Screening of Resistance via Shikimic Acid Accumulation 

Ten plants from the *A. hybridus* population were used for a fast screening using shikimic acid accumulation as parameter to separate resistant plants from susceptible ones. Three disks (4 mm in diameter) of fresh tissue from the youngest expanded leaf per plant were transferred to 2-mL Eppendorf tubes. Shikimic acid accumulation was determined according to Shaner et al. [11] at a single concentration of glyphosate (1000 µM). Assays were repeated three times. Results were expressed in mg of shikimic acid g^−1^ fresh tissue. Plants were separated by high (GSH) or low (GRH) accumulation of shikimic acid, transplanted into pots (30 × 60 cm), and left to grow until maturity to produce new seeds (F_1_) which were used for all future experiments. 

### 4.3. Species Identification

Genomic DNA from ten individuals of each F_1_
*A. hybridus* population was purified using the Qiagen DNeasy Plant mini kit (Qiagen, Valencia, CA, USA) according to manufacturer’s instructions. Species identification was performed by PCR using the specific primers AW473/AW483 (1623 bp) developed by Wright et al. [27]. PCR products were analyzed by agarose gel electrophoresis. 

### 4.4. Dose–response Assays

F_1_ seeds of the GRH and GSH populations were germinated using the same medium as that used in the plant material section. Plants of each *A. hybridus* population were treated using 8 doses (10 plants dose^-1^) of glyphosate (Roundup Energy^®^ SL, 450 g ae L^−1^ as isopropylamine salt, Monsanto Agricultura Española, Madrid, Spain), including 0, 31.25, 62.50, 125, 250, 500, 1000, 2000, and 4000 g ae ha^-1^. Plants were treated at the 4-leaf growth stage, and glyphosate was applied using a spray chamber (SBS-060 De Vries Manufacturing, Hollandale, MN, USA) equipped with 8002 flat fan nozzles delivering 200 L ha^−1^ at the height of 50 cm from plant level. Plant mortality (LD) and fresh weight reduction (GR) were determined at 28 d after treatment (DAT). Data was expressed as percentages in relation to the untreated controls.

### 4.5. ^14^C-Glyphosate Absorption and Translocation

*Amaranthus hybridus* plants of the GRH and GSH populations were treated with ^14^C-glyphosate [glycine-2-^14^C] (specific activity 273.8 MBq mmol^−1^, American Radiolabeled Chemicals, Inc., Saint Louis, MO, USA) + commercial glyphosate. The final glyphosate concentration corresponded to 300 g ae ha^−1^ in 200 L ha^−1^ with a specific activity of 0.834 kBq µL^−1^. GRH and GSH plants at the 4-leaf growth stage were treated with a 1-µL drop (0.834 KBq plant^−1^) placed with a micropipette (LabMate) on the adaxial surface of the second leaf. The plants were handled according to Alcántara-de la Cruz et al. [13] at 12, 24, 48, 72, and 96 h after treatment (HAT) (five plants per population at a time were evaluated in a completely random design). Samples (section of divided plants), once stored into cellulose cones and dried at 60 °C, were combusted in a biological oxidizer (TriCarb 307, Packard Instrument Co., Downers Grove, IL, USA). The ^14^CO_2_ released by combustion was trapped and mixed with 18 mL of Carbo-Sorb® E and Permaf1uor® (Perkin-Elmer, Groningen, Netherlands) (1:1 v/v). Radioactivity of ^14^C was quantified by liquid scintillation spectrometry in a scintillation counter (Beckman LS-6500, Beckman Coulter Inc., Fullerton, CA, USA). Radioactive data was used to calculate the percentages of ^14^C recovered, absorbed and translocated.

### 4.6. ^14^C-Glyphosate Visualization

Plants of GRH and GSH populations at the 4-leaf growth stage were treated and handled at 96 HAT using the same media as those used in the previous section. After removing the nonabsorbed ^14^C-glyphosate by washing the treated leaf three times with water-acetone (1:1, *v*/*v*), the whole plant was removed from the individual pot and its roots carefully washed with distilled water. Excess moisture was removed with paper towel and plants were fixed on filter paper (25 × 12.5 cm) and dried at room temperature (24 °C) for a week. Later, plant samples were placed on phosphor storage film for 4 h and scanned using a Cyclone (phosphor imager, Perkin-Elmer, Bioscience BV Packard, Groningen, Netherlands) to visualize the ^14^C distribution (as glyphosate and/or its potential metabolites) within plants. The assay was carried out with three plants per *A. hybridus* population.

### 4.7. Glyphosate Metabolism

A set of five plants (3–4 leaves) per *A. hybridus* population were treated at 360 g ae ha^−1^ glyphosate as in dose–response assays, while another group of five plants were assigned as the control. Treated and untreated plants were harvested at 48 and 96 HAT and leaf tissue was washed with distilled water, flash-frozen in liquid nitrogen, and stored at −40 °C until use. Glyphosate and its metabolites (AMPA, glyoxylate, sarcosine and formaldehyde) were quantified using a 3D Capillary Electrophoresis Agilent G1600A instrument equipped with a diode array detector (DAD, wavelength range: 190 to 600 nm) [46]. Calibration curves were obtained by using known concentrations of standards (glyphosate and metabolites) supplied by Sigma Aldrich (St. Louis, MI, USA). The experiment had a completely randomized design and repeated three times. Data was expressed as percentages from the total of glyphosate plus metabolites recovered.

### 4.8. EPSPS Enzyme Activity Assays

Five grams (g) of leaf tissue from each *A. hybridus* population finely powdered with liquid nitrogen. EPSPS enzyme extraction was performed following the protocol described by Dayan et al. [47] The total soluble protein (TPS) in the extract (EPSPS basal activity in absence of glyphosate) was determined by the Bradford assay [48], using a Kit for Protein Determination (Sigma-Aldrich, Madrid, Spain) following manufacturer’s instructions. The specific EPSPS activity was assayed in the presence of glyphosate (0, 0.1, 1, 10 100, and 1000 µM) using the EnzChek Phosphate Assay Kit (Invitrogen, Carlsbad, CA, USA) to determine the EPSPS inhibition by 50% (I_50_). Five replications of each population per glyphosate concentration were assayed. EPSPS inhibition was expressed as a percentage relative to the control (absence of glyphosate)

### 4.9. EPSPS Copy Number and Gene Expression

Young leaf tissue of ten individuals from each *A. hybridus* population was collected by taking two samples per plant, one for the total RNA extraction and the other for the genomic DNA extraction. Total RNA was extracted using the Tri Reagent solution (Molecular Research Center, Inc. Cincinnati, OH, USA) according to the manufacturer’s instructions. RNA was treated using the RNase free DNase set (Qiagen, Valencia, CA, USA). First strand complementary DNA (cDNA) synthesis was carried out with 3 µg of total RNA per sample using an M-MLV reverse transcriptase (Promega, Madison, WI, USA) and random hexamers as primers. Cycle conditions: 37 °C for 1 h, 42 °C for 30 min, 50 °C for 10 min, and 15 °C for 10 min. The gDNA was purified using the Qiagen DNeasy Plant mini kit (Qiagen, Valencia, CA, USA) according to manufacturer’s instructions.

Copy number (from gDNA) and gene expression (from cDNA) assays were performed using EPSPS and ALS primer pairs developed by Gaines et al. [10]. Reactions were performed using a qRT-PCR Bio-Rad CFX connect thermal cycler and the following amplification profile; 50 °C for 2 min, 95 °C for 10 min, 40 cycles at 95 °C for 15 s and 60 °C for 1 min, and 95 °C for 15 s. PCR reactions were set up in 20 µL of SYBR Green PCR Master Mix (Bio-Rad, Hercules, CA, USA), following manufacturer’s instructions. Controls containing water were included to check for contamination in the reaction components. The *ALS* gene was used as a reference gene to normalize qRT-PCR results. Reactions was carried out in triplicate per sample and standard curves were performed for each primer pair to confirm the amplification efficiency (E = 100 ± 10%). The relative expression levels were calculated from the threshold cycle (Ct) values and the primer efficiencies by the Pfaffl method [49]. The *EPSPS* gene copy number was determined [10]. Results were expressed as relative *EPSPS* gene copies in relation to the *ALS* gene [49].

### 4.10. Mutations in the EPSPS Coding Sequence

Sanger sequencing was used to detect target site mutations in the *A. hybridus EPSPS* gene sequence using the DNA purified previously. For the sequencing reactions, a 196 bp long DNA fragment was amplified using the primers described in Lorentz et al. [25]. The PCR conditions were 15 min of preincubation at 95 °C, followed by 45 cycles at 94 °C denaturating step (30 s), 55 °C annealing temperature (40 s), 70 °C elongation (40 s) and a final extension step for 10 min at 70 °C (Bio-Rad T100 thermal cycler, Hercules, CA, USA). Each plant (ten per population) was sequenced in triplicate.

### 4.11. Structural Modeling

The spatial structure of the wild type (WT; the GSH population) isoform of the EPSPS was reconstructed by a homology modeling approach using the swiss-Model software [50]. The previously obtained spatial structure EPSPS from Vibrio cholerae (PDB code: 3nvs) was used as a template for *A. hybridus* EPSPS reconstruction. To generate the promodels of mutant *A. hybridus* EPSPS isoforms, the appropriate amino acids at the positions 102 (Thr-Ile), 103 (Ala-Val) and 106 (Pro-Ser) were changed using the DeepView software version 4.1.0. This same software was used to analyze structural differences between the WT and mutant isoform.

### 4.12. Statistical Analysis

The amount of glyphosate causing the fresh weight reduction (GR_50_), plant mortality (LD_50_) and EPSPS inhibition (I_50_) by 50% of each *A. hybridus* population were determined using the following log-logistic equations [51]: Y = d/1 + (x/g)^b^ (three-parameters for GR_50_ and LD_50_) and Y = c + {(d-c)/[1 + (x/g)^b^]} (four-parameters for I_50_): where Y is the percentage of fresh weight, mortality and/or EPSPS inhibition relative to the control; c and d are the lower and upper limits, respectively, of the curve; b is the slope of the curve; g = herbicide concentration at the inflection point (i.e., LD_50_, GR_50_, or I_50_), and x is the glyphosate dose. The three-parameters model assumes that the lower limit is zero. Regression analyses were conducted using the *drc* package with program R 3.2.5, and plotted using SigmaPlot 11.0 (Systat Software, Inc., San Jose, CA. USA).The resistance indexes (RI = R/S) were computed as R-to-S ratios.

Data regarding basal EPSPS activity, absorption, translocation, and metabolism were subjected to ANOVA using Statistix 9.0 (Analytical Software, Tallahassee, FL, USA). Model assumptions of normal distribution of errors and homogeneous variance were graphically inspected. Differences of *p* < 0.05 were considered significant and Tukey’s test was conducted for means comparison.

## 5. Conclusions

The GRH *A. hybridus* population presented reduced translocation of glyphosate and signals of *EPSPS* gene amplification; however, the contribution of these mechanisms in the high resistance level showed by this population was minimal. That resistance was due to the triple amino acid substitution from TAP (wild type) to IVS (triple mutant) (Thr102Ile + Ala103Val + Pro106Ser), occurring in the conserved region of the *EPSPS* gene, found in the GRH *A. hybridus* population. This novel triple substitution increased the hydrogen-bond distance between the Gly-101 and Arg-105 positions restricting the binding of glyphosate with the EPSPS enzyme.

## Figures and Tables

**Figure 1 ijms-20-02396-f001:**
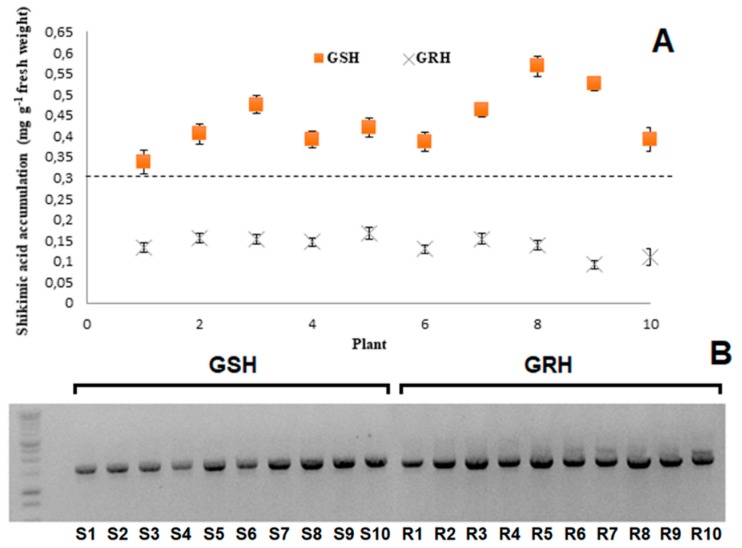
(**A**) Shikimic acid accumulation in *A. hybridus* plants, from Cordoba, Argentina, and its classification in resistant (GRH) or susceptible (GSH). (**B**) Gel image of PCR with species-specific primers. Lane 1 is a 1 kb ladder with bands of 10, 8, 6, 5, 4, 3, 2.5, 2, 1.5, 1, 0.7, 0.5, and 0.3 kb. Lanes 2 to 11 are the ten individuals of the GSH population, and lanes 12 to 21 are the ten GRH individuals.

**Figure 2 ijms-20-02396-f002:**
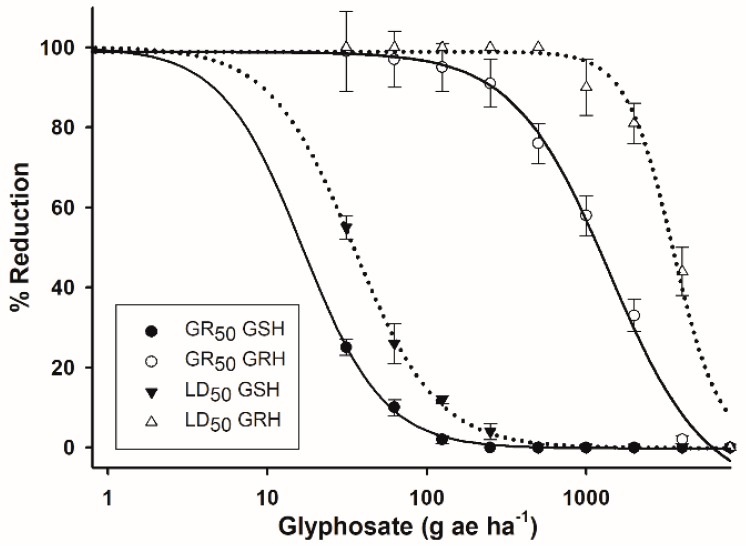
Log-logistic curves of the GRH and GSH *A. hybridus* populations from Cordoba, Argentina, evaluated 28 d after treatment. Solid lines are the dose–response curves with respect to percentage of fresh weight reduction (GR_50_). Dotted lines are the dose–response curves with respect to percentage of plant survival (LD_50_). Vertical bars represent the standard error of the mean (*n* = 10).

**Figure 3 ijms-20-02396-f003:**
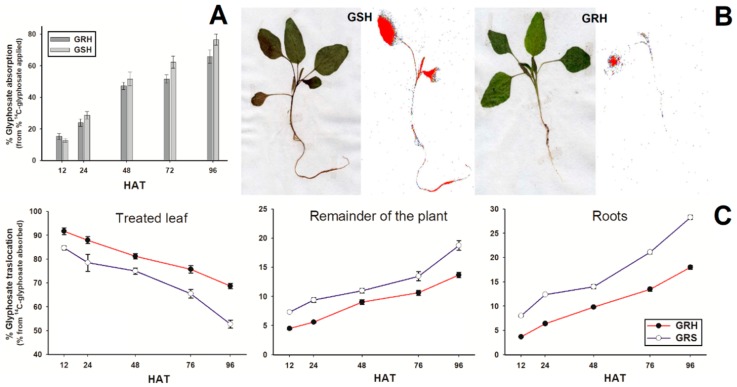
^14^C-glyphosate absorption and translocation in GRH and GSH *A. hybridus* populations from Cordoba, Argentina, at different h after treatment (HAT). (**A**) Percentage of ^14^C-glyphosate absorbed from total applied. Vertical bars represent the standard error of the mean (*n* = 5). (**B**) Digital and autoradiograph images of ^14^C-glyphosate (or metabolites) distribution within GSH and GRH plants. The red color indicates a high concentration of ^14^C-herbicide. (**C**) Translocation of ^14^C-glyphosate from treated leaf to remainder of plants and roots. Vertical bars represent ± standard errors (*n* = 5).

**Figure 4 ijms-20-02396-f004:**
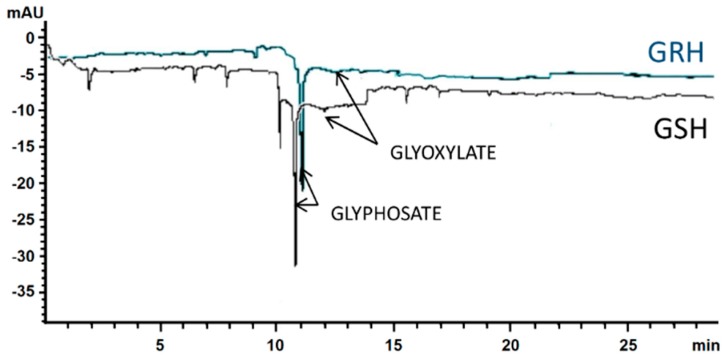
Electropherograms of glyphosate metabolism in GRH (blue) and GSH (black) plants of *A. hybridus* from Cordoba, Argentina, at 96 h after treatment with glyphosate (300 g ae ha^−1^).

**Figure 5 ijms-20-02396-f005:**
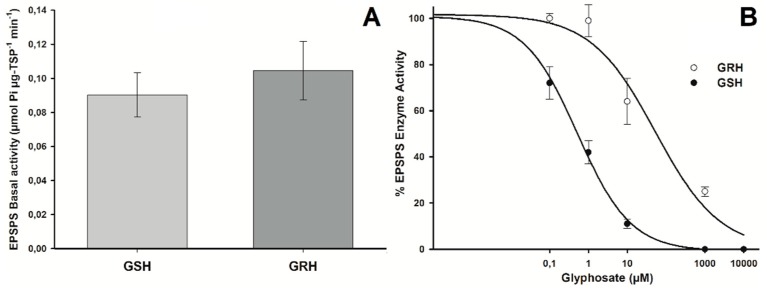
EPSPS activity in GRH and GSH plants of *A. hybridus* from Cordoba, Argentina. (**A**) Basal EPSPS activity (absence of glyphosate). Histograms represent the means and vertical bars the standard error (*n* = 5). (**B**) Dose–response curves of the EPSPS enzyme activity, expressed as percentage of the untreated control, exposed to different glyphosate concentrations (µM).

**Figure 6 ijms-20-02396-f006:**
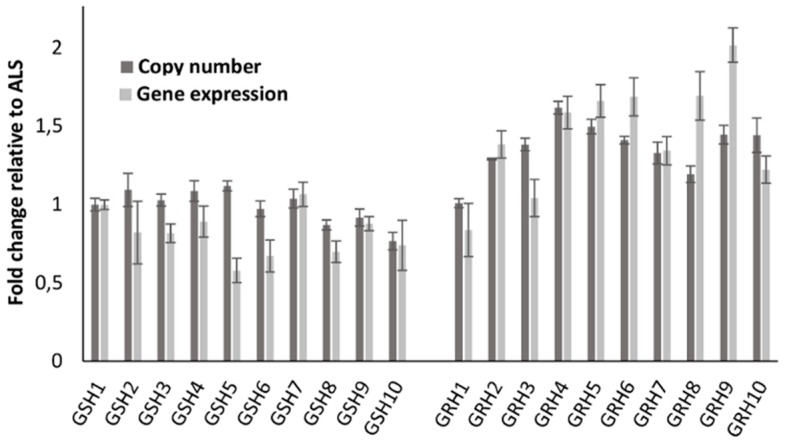
*EPSPS* gene copy number and expression level relative to the ALS gene in ten individuals of the GRH and GSH *A. hybridus* populations from Cordoba, Argentina. Values represent the average of three technical replicates per plant.

**Figure 7 ijms-20-02396-f007:**
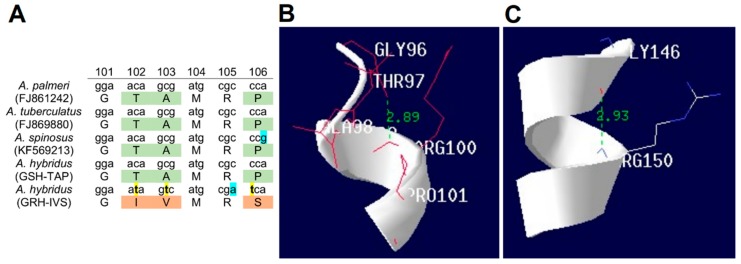
Comparison of *EPSPS* gene sequences and EPSPS ribbon diagrams of the GRH and GSH *A. hybridus* populations from Cordoba, Argentina. (**A**) Partial alignment of nucleotides and predicted amino acids of the *EPSPS* genes. Amino acid position based on the start codon (ATG) of *Arabidopsis thaliana* (GenBank: CAA29828.1) *EPSPS* gene sequence. Nucleotides highlighted in blue do not result in amino acid substitution. Nucleotides highlighted in yellow result in amino acid substitution. Orange boxes highlight a triple amino acid substitution from TAP (glyphosate susceptible plants or wild type) to IVS (glyphosate-resistant plants). Ribbon diagram of EPSPS target site (101–106 positions) from GSH plants (**B**) and GRH (**C**) plants. In green: H-bond distances between Gly-101 and Arg-105.

**Table 1 ijms-20-02396-t001:** Parameters of the log-logistic equations^a^ used to calculate the glyphosate concentrations required to kill (LD_50_), reduce the fresh weight (GR_50_) and inhibit the EPSPS activity (I_50_) by 50% in *A. hybridus* populations from Cordoba, Argentina.

Population	c	d	b	Concentration	RI	*p-value*
LD_50_ (g ae ha^−1^) ^†^						
GRH	-	98.9	0.21	3503.4 ± 34.7	100.6	< 0.0001
GSH	-	100.1	0.14	34.8 ± 2.8	--	< 0.0001
GR_50_ (g ae ha^−1^) ^†^						
GRH	-	98.9	11.37	1395.2 ± 164.5	83.9	< 0.0001
GSH	-	100.0	0.26	16.6 ± 1.7	--	< 0.0001
I_50_ (µM) ^‡^						
GRH	0.10	101.7	0.52	52.8 ± 3.4	100.9	< 0.0001
GSH	0.84	100.9	0.63	0.52 ± 0.07	--	< 0.0001

^a^ Y= d/1 + (x/g)^b^ (three-parameters)^†^ or Y = c + {(d-c)/[1 + (x/g)^b^]} (four-parameters)^‡^: where Y = response in relation to the control, c = lower limit, d = upper limit, b = slope of the curve, g = herbicide concentration at the inflection point (i.e., LD_50_, GR_50_, or I_50_), and x = herbicide concentration. The model of three-parameters assumes that the lower limit is zero. RI= Resistance indexes (R/S) computed as R-to-S LD_50_, GR_50_ or I_50_ ratios. ± Standard error of the mean (*n* = 10 for LD_50_ and GR_50_, and *n* = 5 for I_50_).

**Table 2 ijms-20-02396-t002:** Glyphosate metabolism (glyphosate/metabolites in nmols g^−1^ weight fresh) in *A. hybridus* plants at 48 and 96 h after treatment (HAT) treated with glyphosate at 300 g ae ha^−1^.

Population	HAT	Aerial Part		Roots	
		Glyphosate	Metabolites	Glyphosate	Metabolites
GRH	48	436.9 ± 2.5	ND	47.9 ± 4.6	ND
	96	494.3 ± 3.7	ND	114.3 ± 5.0	ND
GSH	48	522.9 ± 8.4	ND	83.5 ± 7.0	ND
	96	605.0 ± 12.5	ND	237.5 ± 7.2	ND

ND (not detected).

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
