# Peer review of "The Triple Amino Acid Substitution TAP-IVS in the EPSPS Gene Confers High Glyphosate Resistance to the Superweed Amaranthus hybridus"

_ijms, 2019, doi:10.3390/ijms20102396_

Round 1

Reviewer 1 Report

I read this manuscript with high interest. This novel triple mutation is very exciting. This is particularly more so with the discovery of another TAP-IVS from the same species originating also from Cordoba, Argentina. Both populations in general display high level glyphosate resistance. I would like to congratulate the authors for the hard work in characterizing this mechanism and a well-written manuscript. Please see and address my comments below.

Title: The title is on-point. However, considering the recent publication of Perotti et al. (2018), I feel the weight of the word 'novel' in the title has been diminished. Would the authors consider revising the title?

Abstract 

Line 20-21: 'Inadequate adoption of this technology..'. I am a bit confused with this sentence. Were the authors referring to farmers not taking up GR crops, or the increased usage and dependence on glyphosate as a result of GR crops adoption? The sentence seems to imply the former, which I don't think is what the authors are saying. Please revise.

Line 31: structural modelling?

Line 31-33: Based on the results, reduced/impaired translocation, together with the TAP-IVS mutation, contributed to glyphosate resistance in this population. I suggest including reduced/impaired translocation as a factor in the abstract.

Introduction

Line 58: discard the comma after Amaranthus.

Line 60: I suggest replacing ‘develop’ with ‘evolve’

Materials and Methods

Line 78: in the summer season of 2015/2016.

Line 82-83: Please include the GPS coordinates of the GSH population.

Line 87 - 88: ...and kept in the growth chamber until herbicide applications. Afterwards, treated plants...

Line 107: dose-1

Line 105-112: Was this experiment repeated? Were all 10 plants in a pot? How many replicate pots?

Line 114: …were treated with 14C-glyphosate…

Line 127: treated leaf

Line 132: ‘…(as glyphosate and its potential) within plants.’ Can the authors explain what they meant by glyphosate and its potential?

Line 144: discard ‘of’

Line 146: Five gram (g) of leaf tissue from each A. hybridus population was finely powdered…

Line 156: …was collected by taking…

Line 183: Were there ten biological and three technical replicates for each GSH and GRH populations? Please clarify.

Line 188: …A. hybridus EPSPS reconstruction.

Line 195-199: Please check the consistency between this part and line 232-236 and make it clearer. For example, in line 198, b is the slope of the inflection point (i.e., GR50, LD50 or I50) but in line 233, b is slope of the curve and g is the herbicide concentration at the inflection point (i.e., GR50, LD50 or I50). There is no definition for g in line 195-199.

Results

Figure 1: I suggest the authors include lane numbers under the gel image in Figure 1B.

Line 220: Is 960 g ae ha-1 the rate used by Argentinian farmers? Is that the recommended label rate? Could the authors include it (i.e., 960 g ae ha-1) in this sentence?

Line 232-236: Again, please check the consistency between this part and line 195-199.

Line 268: ND (non-detectable or not detected)?

Line 294: …A. hybridus EPSPS cDNA…

Line 307: hydrogen bond (H-bond)

Discussion

Line 320-321: Could the authors discuss the difference in GR50 levels between this study and the ones in Perotti et al.?

Line 325-333: I agree with the authors especially when we need to consider weed management aspects in response to multiple resistance evolution. However, I could not see the relevance of this section to this paper, which is looking at the TAP-IVS mutation and glyphosate resistance. Furthermore, there was no investigation into possible herbicide resistance to ALS inhibitors. Thus, it seems rather out of place in this situation.

Line 334-342: The reduced/impaired translocation in the GRH population is small but significant, based on the results. The authors could possibly highlight this along with the TAP-IVS mutation/substitution, especially in the title, giving this manuscript a different dimension or novelty in contrast to the paper by Perotti et al. (2018). This is just a suggestion and is entirely up to the authors to decide.

Line 334-336: Please re-structure this sentence. I suggest giving the impaired translocation its own sentence.  

Line 341-342: …cannot be explained by this mechanism alone, suggesting…

Line 342: TSR mechanism

Line 379-386: How does the alanine side chain occupying more space than the valine one contributes to increase glyphosate resistance? Could the authors please elaborate?

Line 387: This sentence is unnecessary

Conclusions

All good!

Author Response

Title: The title is on-point. However, considering the recent publication of Perotti et al. (2018), I feel the weight of the word 'novel' in the title has been diminished. Would the authors consider revising the title? Title was slightly modified as suggested (L2-4)

Abstract 

Line 20-21: 'Inadequate adoption of this technology..'. I am a bit confused with this sentence. Were the authors referring to farmers not taking up GR crops, or the increased usage and dependence on glyphosate as a result of GR crops adoption? The sentence seems to imply the former, which I don't think is what the authors are saying. Please revise. DONE (L21)

Line 31: structural modelling? DONE (L34)

Line 31-33: Based on the results, reduced/impaired translocation, together with the TAP-IVS mutation, contributed to glyphosate resistance in this population. I suggest including reduced/impaired translocation as a factor in the abstract. DONE (L34-36). In addition, a statement was added in L28-29 highlighting the translocation result.

Introduction

Line 58: discard the comma after Amaranthus. DONE (L63)

Line 60: I suggest replacing ‘develop’ with ‘evolve’, DONE, but used “select” (L65)

 Materials and Methods

Line 78: in the summer season of 2015/2016. ????

Line 82-83: Please include the GPS coordinates of the GSH population.

Line 87 - 88: ...and kept in the growth chamber until herbicide applications. Afterwards, treated plants... DONE (L92-94)

Line 107: dose-1 DONE (L112)

Line 105-112: Was this experiment repeated? Were all 10 plants in a pot? How many replicate pots? This experiment was not repeated because the F1 seeds, which were previously selected in the shikimic acid assay, were used for all the experiments in this research (all carried out in the UCO-Spain). In addition, these populations had already been characterized as resistant and susceptible to glyphosate previously in Argentina.

Line 114: …were treated with 14C-glyphosate… DONE (L119)

Line 127: treated leaf DONE (L132)

Line 132: ‘…(as glyphosate and its potential) within plants.’ Can the authors explain what they meant by glyphosate and its potential? DONE (L137)

Line 144: discard ‘of’ DELETED

Line 146: Five gram (g) of leaf tissue from each A. hybridus population was finely powdered… DONE (L151)

Line 156: …was collected by taking… DONE (L161)

Line 183: Were there ten biological and three technical replicates for each GSH and GRH populations? Please clarify. DONE (L188)

Line 188: …A. hybridus EPSPS reconstruction. DONE (L193)

Line 195-199: Please check the consistency between this part and line 232-236 and make it clearer. For example, in line 198, b is the slope of the inflection point (i.e., GR50, LD50 or I50) but in line 233, b is slope of the curve and g is the herbicide concentration at the inflection point (i.e., GR50, LD50 or I50). There is no definition for g in line 195-199. DONE (L203)

 Results

Figure 1: I suggest the authors include lane numbers under the gel image in Figure 1B. DONE

Line 220: Is 960 g ae ha-1 the rate used by Argentinian farmers? Is that the recommended label rate? Could the authors include it (i.e., 960 g ae ha-1) in this sentence? DONE (L226)

Line 232-236: Again, please check the consistency between this part and line 195-199. DONE (L238-242)

Line 268: ND (non-detectable or not detected)? DONE (L272)

Line 294: …A. hybridus EPSPS cDNA… DONE (L298)

Line 307: hydrogen bond (H-bond) DONE (L310)

 Discussion

Line 320-321: Could the authors discuss the difference in GR50 levels between this study and the ones in Perotti et al.? DONE (L325-334)

Line 325-333: I agree with the authors especially when we need to consider weed management aspects in response to multiple resistance evolution. However, I could not see the relevance of this section to this paper, which is looking at the TAP-IVS mutation and glyphosate resistance. Furthermore, there was no investigation into possible herbicide resistance to ALS inhibitors. Thus, it seems rather out of place in this situation. DELETED

Line 334-342: The reduced/impaired translocation in the GRH population is small but significant, based on the results. The authors could possibly highlight this along with the TAP-IVS mutation/substitution, especially in the title, giving this manuscript a different dimension or novelty in contrast to the paper by Perotti et al. (2018). This is just a suggestion and is entirely up to the authors to decide. Title was changed according to the first reviewer comments

Line 334-336: Please re-structure this sentence. I suggest giving the impaired translocation its own sentence.  DONE (L337-339)

Line 341-342: …cannot be explained by this mechanism alone, suggesting…DONE (L344)

Line 342: TSR mechanism. DONE (L345)

Line 379-386: How does the alanine side chain occupying more space than the valine one contributes to increase glyphosate resistance? Could the authors please elaborate? Our statement is based on what was previously explained in L373-381

Line 387: This sentence is unnecessary. DELETED

 Conclusions

All good!

Reviewer 2 Report

The manuscript is very well written; clear, precise, and easy to understand. The title clearly reflect the contents of the paper the abstract is sufficiently informative, material and methods are detailed and results are properly discussed.

Below is a specific point.

As I read the manuscript, I consider the remarks about weed resistance to be incomplete. Indeed, in my opinion even if it is a paper with a strong emphasis on molecular genetics and biology may be useful to remind in the text of the introduction section that a solution to avoid the selection of weed resistance, due to the use of GR-crops or to the use of lower than recommended herbicide doses, might be to add agronomic practices such as the use of rotations with different crop species characterized by a different weed management technique as stated by Deligios et al 2018 and 2019 (https://doi.org/10.3390/su11061653 and https://doi.org/10.3390/su10072258).

Author Response

All changes suggested by reviewer 1 were followed, however, we consider that it is not appropriate to emphasize the weed management measures in this work. Even, part of the previous version was deleted, although interesting, it did not add to this work.

All changes suggested by reviewer 1 were followed, however, we consider that it is not appropriate to emphasize the weed management measures in this work. Even, part of the previous version was deleted, although interesting, it did not add to this work.

We appreciate your valuable revsion